

# Long-term evaluation of the Sub-seasonal to Seasonal (S2S) dataset and derived hydrological forecasts at the catchment scale

Marianne Brum[1], Dirk Schwanenberg[1]

[1]KISTERS AG, BU Water, 52076 Aachen, Germany

*Correspondence to*: Marianne Brum (marianne.brum@kisters.de)

**Abstract.** Recently, projects such as the S2S (Sub-seasonal to Seasonal) have surfaced with the goal of investigating the potential benefits of operational applications of medium- to long-term weather forecasts from two weeks to three months. Key challenges are to quantify forecast uncertainty and verify these predictions considering the downstream users. This work evaluates the meteorological lead-time performance and 5-years skill evolution of nine models of the S2S project alongside
discharge predictions from a coupled hydrological model. Moreover, an analysis of the predictors of Numerical Weather Prediction (NWP) quality and an evaluation of the correlation between meteorological and hydrological quality improvement over time is carried out. Results show that the S2S models have skill at the catchment-scale, particularly for lower threshold levels, and that ensemble size is the main predictor of NWP performance. Discharge simulations forced with S2S predictions remain skilful up to one month. The quality of the S2S has increased over time, and there is a strong correlation between
meteorological and hydrological improvements. We conclude that S2S products may provide added value to end-users of water resources applications.

## 1 Introduction

Many water resources systems rely on hydrological forecasts for the efficient management of their assets, which themselves depend on Numerical Weather Predictions (NWP) as forcing. That includes water resources applications for navigation,
hydropower, flood and drought management, water supply and irrigation. These systems can thus potentially achieve higher operational efficiency and reliability through improvements in the NWPs lead-time accuracy and forecast horizon. The longer the horizon of skilful forecasts, the earlier can decision-makers take proper precautions against natural hazards such as flooding and drought, the more economical gain can energy and water supply operators achieve through efficient management of reservoirs, and the more reliable is the provision of water for agricultural and domestic usage.

In that regard, recently projects such as the S2S (Sub-seasonal to Seasonal) have surfaced with the goal of investigating the potential benefits of operational applications of medium- to long-term forecasts from two weeks to three months. One of the main challenges remains the quantification of uncertainty and verification of these NWPs and the communication of results to increase the confidence of downstream end-users (Vitart, Robertson and Anderson, 2012). Though the performance of hydrological forecasts is heavily influenced by the uncertainty in the NWPs (e.g. Cuo et al., 2011), improvements in the



hydrological horizon of skilful predictability are not directly proportional to gains in the forcing's lead-time accuracy due to the complex and non-linear nature of hydrological processes (Pappenberger et al., 2011). Therefore, to verify the true benefit of S2S NWPs to end-users, both the meteorological forecasts and the response generated from the hydrological models must be evaluated.

Several recent studies have verified hydrometeorological forecast systems for short to medium prediction ranges (e.g.
Anderson et al., 2019; Fan et al., 2015; Velázquez et al., 2009; Yucel et al., 2015), while others have evaluated the evolution in performance of numerical weather predictions in the same range (e.g. Böhme et al., 2011; Mass et al., 2002; Rodwell et al., 2010). Generally, a positive correlation between number of ensemble members, model resolution and forecast skill has been found. On the sub-seasonal to long-term range, only a small number of papers have evaluated the quality of the meteorological predictions generated by NWPs (e.g. DeFlorio et al., 2019; He et al., 2020; Olaniyan et al., 2018; Wang and Robertson, 2019).
Furthermore, few have assessed the coupled hydrometeorological performance in this prediction range. Notably, Quedi and Fan (2020) verified the ECMWF model included in the S2S project in the context of a large catchment which is relevant to hydropower generation in Brazil. Through a set of different metrics, the authors concluded that the S2S prediction can be useful for reservoir inflow forecasting and applicable in the planning of operations at the monthly range. Li et al. (2019) evaluated eight of the eleven available S2S models coupled with hydrological forecasts for four sub-basins in south China and
reached similar findings regarding the applicability of the sub-seasonal predictions to hydrometeorological forecasting systems, though they noted that bias correction might be necessary. Thus, more work is necessary to verify the added value of sub-seasonal to seasonal forecasts to water resources applications.

Notably, advances in model quality through increased model resolution, coverage, couplings, and implementation of additional atmospheric processes must be balanced with storage capacity, computation power and availability of results in view of their
end-uses. However, few publications have investigated the variation of S2S NWPs quality over time in the context of their downstream usage. More research is therefore needed to quantify the improvement of sub-seasonal forecasts considering their downstream applications, particularly regarding the forecast horizon, to direct outlooks in research, maintenance, and development of forecasting systems.

In this work, we hypothesize that the hydrological forecast quality at the sub-seasonal range has improved over time, and that
those improvements correlate with improvements in the S2S meteorological forecasts. Moreover, we expect some commonly cited model features to be definite model predictors, namely the size of the ensemble and the spatial model resolution. In that context, these hydrological forecasts should outperform or at least perform as well as base models commonly used in an operational setting, such as persistence and the annual discharge cycle. We verify over five years of precipitation and temperature predictions from nine NWPs of the S2S dataset alongside forecasts from an integrated semi-distributed
hydrological model. We apply a set of both probabilistic and deterministic verification metrics and select two meteorological and four hydrological benchmarks for comparison. The lead-time dependent quality as well as the evolution of skill of meteorological and hydrological predictions is verified. Moreover, we carry out an analysis of the predictors of sub-seasonal ensemble model performance and of the correlation between meteorological and hydrological quality variation over time. This





paper aims to contribute to the existing literature on sub-seasonal hydrometeorological evaluation, as well as providing an
overview of the added value of the lead-time performance evolution of S2S NWPs in downstream hydrological applications
such as early-warning systems, reservoir and navigation planning, and monthly to seasonal decision making.

## 3 Materials and Methods

### 3.1 Pilot Areas

We select two highly gauged sites for evaluation (Figure 1). The first is the Upper Main river catchment in Germany, which
is a tributary of the Rhine River. Two sub-catchments are defined according to two streamflow gauging stations at Schwürbitz
and Kemmern following the work of Montero et al. (2016), where the authors implemented a hydrological model for the
region. Each has a drainage area of 2,328.3 km² (Kemmern) and 1,799.5 km² (Schwürbitz) and together they encompass four
grid points of the S2S raster grid. The daily mean discharge at the outlet stations was 30.2 m³/s from 1941 to 2020 (Schwürbitz)
and 44.3 m³/s from 1931 to 2020 (Kemmern) (BLfU, 2021). Mean precipitation in the catchment ranges from 600 to 1,100
mm/year and average yearly temperatures vary from 6°C to 9°C (derived from Montero et al., 2016).

This study also evaluates meteorological data in the federal state of Rhineland-Palatinate (*Rhineland-Pfalz*, RLP), located in
the western part of Germany at the border to Belgium, Luxembourg and France. The region has a total area of 19,854.2 km²
(RPKK, 2021) with varying mean precipitation rates from 400 to 1,400 mm/year and average yearly temperatures from 6°C
to 11°C depending on the elevation. The RLP Environmental Agency (2020) subdivides the area into 35 main river sub-
catchments, as shown in Figure 1. However, the whole RLP contains only 5 points of the S2S grid, thus it is not meaningful
to analyse each catchment. Instead, we verify the whole RLP as a single area. Though not very operationally interesting, we
deem the experiment relevant because coarse global products are often used as boundaries for finer, regional models and as an
alternative computationally cheaper source of information when employing high-resolution products is not feasible. As such,
only the meteorological performance of the S2S is evaluated for RLP.

### 3.2 Data

The S2S (Sub-seasonal to Seasonal) project results from a global research initiative that aims to fill the gap between short- and
medium- range forecasts (up to two weeks) and seasonal predictions (Vitart and Robertson, 2018). The S2S dataset is
composed of 11 ensemble prediction models with varying spatial resolution, number of ensemble members, issuance
frequency, and time range, from 32 to 62 days maximum lead-time. The models are maintained by various meteorological
centres throughout the world and hosted in an upscaled common 1.5 x 1.5 degrees regular grid at the ECMWF data portal since
2015. Some models started being provided later, thus have a slightly smaller temporal coverage.

Daily forecasts for precipitation, minimum and maximum temperature are downloaded from 2015 to 2020 for the nine selected
models of the S2S dataset (Table 1). Please note that the control members (not perturbed) are not included here. This diverse





dataset allows for comparison of multiple models that have different forcing and configurations: aside temporal gaps in model
availability, 234 ensembles are verifiable at every grid point and timestep.

We extracted precipitation observations from the *Deutscher Wetterdienst* (DWD) REGNIE (*REGionalisierte NIEderschläge*) dataset, which consists of daily values of precipitation of more than 2,000 stations in Germany interpolated to a 1 x 1 km grid (Rauthe et al., 2013). Temperature measurements are obtained from 76 well-distributed monitoring stations in the catchments of interest, provided by the DWD at the Climate Data Center OpenData portal (DWD, 2020). Daily discharge observations for
the outflows of the two Upper Main sub-catchment outlets (Schwürbitz and Kemmern stations) are downloaded from the Gewässerkundlicher Dienst Bayern, made available by the Bayerisches Landesamt für Umwelt (BLfU, 2021).

### 3.3 Methods

### 3.3.1 Evaluation Procedure

First, we correct the temperature value of gridded products by elevation making use of the ETOPO5 global-elevation map
from the U.S.A. National Centres for Environmental Information, NCEP (Mitchell, 2014), while station-based measurements are adjusted according to each station elevation. Next, we spatially aggregate precipitation and temperature forecasts through an area-weighted averaging proportionally to the sub-catchment area within each raster grid point. Measurements of temperature are averaged over the stations within the evaluated regions.

We then run daily hydrological reforecasts from 2015 to 2020 using the aggregated meteorological predictions as input. One
simulation is run for each weather forecast, model ensemble member and sub-catchment. We compute nine daily lead-time dependent verification metrics, two meteorological and four hydrological benchmarks for the meteorological and hydrological forecasts at each region of interest. Metrics are chosen to assess not only the accuracy of sub-seasonal predictions, but also a range of forecasts properties that are relevant to the end-user (such as consistency). The hydrological modelling and the verification metrics employed are described in further detail in the following sub-sections.

For the performance evolution evaluation, we average the daily scores over a year and assess them against the benchmarks (as skill scores) to account for natural variability. A composite-model (created by combining the scores of each S2S NWP included in this evaluation) is also included to represent the overall performance of the S2S dataset over time. From it, we compute the lead-time gains from 2015 to 2020: i) for any given score, ii) the gain in lead-time is defined as the difference between the lead-time for the score in 2020 and iii) the lead-time for the same score at the beginning of the forecast record.

Moreover, we carry out a correlation analysis with the meteorological performance of the different S2S models to highlight model performance predictors. Scores are averaged for the first two weeks (up to medium range) and for the sub-seasonal range. We consider ensemble size, maximum lead-time, and original model resolution as possible factors. Due to the small sample size (nine models), we bootstrap the correlation with 10,000 resamples with replacement for each factor and score pair and consider an interval of confidence of 95% for significance.





### 3.3.2 Verification Metrics and Benchmarks

Measuring the performance of meteorological and hydrological predictions requires the definition of appropriate quality metrics. In that regard, a series of performance indicators that can quantify the relevant attributes for each use-case are selected as recommended in the literature (Bartholmes et al., 2009). According to Murphy (1993), the three main aspects of a good forecast are consistency, quality (or accuracy), and value. Regarding quality, Murphy and Winkler (1987) proposed it is a combination of attributes: unconditional bias, the systematic difference between forecasts and observations; reliability, the bias between forecasts and observations conditional on the forecasts; discrimination, the same as resolution, but conditional on the observations; and sharpness, the certainty in a prediction (correct or not) (Jolliffe and Stephenson, 2012; Wilks, 2019).

Table 2 below summarizes all nine metrics applied in this study. We use the ensemble mean of each model for the deterministic scores, and the complete ensemble for the others. We verify the implemented scores against the EVS software (Brown, 2010) using a subset of GEFS/R2 reforecasts (Hamill et al., 2013) from 2006 until 2010 and measurements for 11 stations in Germany provided by the CDC's OpenData archive (DWD, 2020). Hereon, only the most relevant results are discussed for simplicity. For an in-depth analysis of the evaluation metrics see Wilks (2019) and Joliffe and Stephenson (2012).

To ensure a robust evaluation of NWPs performance, the absolute scores are compared with a series of benchmarks or base models. The selected baselines for this study follow the recommendations of Pappenberger et al. (2015) on their comprehensive evaluation of benchmarking for hydro-meteorological systems. For meteorological variables, two are employed: persistence, where the last known parameter value is propagated into the future; and advanced climatology, or the long-term annual climatology for each day, here computed for 10 years. For discharge, four are used: meteorological persistence; advanced climatology (or annual discharge cycle); simple persistence, or where the hydrological observations are persisted instead of the forcing; and a perfect forecast, which is a hydrological simulation forced with observed meteorological data.

For limit-based scores, we select thresholds based on the literature (e.g. Bartels et al., 2017; Pappenberger et al., 2011; Randrianasolo et al., 2010). We use Q70 and Q90 for precipitation and temperature, and for discharge Q50, Q70, Q90 and Q97 (or 50%, 30%, 10% and 3% quantiles). We set here the probability threshold of value at 50% and a cost-loss ratio equal to the event climatology to obtain the maximum possible score.

### 3.3.3 Hydrological Modelling

We define a hydrological model based on an implementation by Schwanenberg and Montero (2016) and by Montero et al. (2016) of the HBV-96 conceptual rainfall-runoff model (Lindström et al., 1997). The model for the Upper Main is defined as two sub-catchments with a simple, non-linear retention scheme for routing (Figure 1). Here, the HBV is treated as a semi-distributed model by discretizing the two sub-catchments with outflows at Schwürbitz and Kemmern into 30 sub-basins with individual hydrological characteristics (such as land-use, elevation, and soil type). The meteorological forcing is aggregated across the sub-catchments. We apply the rainfall-runoff routine to each of the 30 sub-basins and generate daily values, which are concentrated at nodes N2 and N3 and then routed to the outflow stations N0 and N1.





In addition, we perform a variational data assimilation (DA) procedure to receive a best guess of the system state at forecast time in each sub-catchment. The first order sensitivities required for this approach are provided through a reversed adjoint model and algorithmic differentiation (Griewank and Walther, 2008) of the original hydrological model. Before hydrological predictions are made, we run the full DA procedure once for 10 years of available observation data at the daily level. Then, every new simulation is initialized with the DA-generated optimal states.


We obtain the optimal parameters for the HBV model from the original implementation by Montero et al. (2016), and calibrated the routing scheme semi-manually through a parameter optimization loop. We reuse the required datasets for calibration and subsequent validation from the daily observations taken from Montero et al. (2016), without overlap. Without DA, the model achieves a Nash-Sutcliffe Efficiency (NSE) of 0.88 for both calibration and validation at gauge Schwürbitz, and 0.90 for calibration and 0.92 for validation at gauge Kemmern. Validation with DA for the last 10 years of available data results in an increased R² of 0.94 at Kemmern and 0.96 at Schwürbitz. The high correspondence to the observed data allows the discharge forecast skill to reflect the improvements in the NWPs with as little interference as possible from the hydrological model.


## 4. Results and Discussion

### 4.1 Meteorological performance of S2S forecasts at the catchment level

Figure 2 shows the site-averaged precipitation BS Q70, BS Q90, CRPS, FCS Q70, MAE and Bias score values per lead-time for all S2S models and meteorological benchmarks. The plots do not show results for temperature because the findings for both variables are alike, albeit temperature has constantly better performance than precipitation. Likewise for RLP and Upper Main, where absolute values are similar and thus are not discretized. With the exception of the FCS (for which there is an improvement), a trend of worsening forecast quality with lead-time up to a limit yield point around 10 to 14 days is evidenced for all scores and all models verified, after which performance becomes approximately constant. Notably, performance after the yield lead-time is sometimes superior to the benchmarks concerning the probabilistic metrics, particularly at lower thresholds. At Q90, five of the nine models have skill over the climatology, while at Q97 those same models are only able to match the benchmark score. Persistence, on the other hand, is shown to be an unreliable baseline for skill discrimination, because


it is always outperformed. For deterministic accuracy measures of precipitation, the advanced climatology beats all models after five to seven days, while for RMSE (not shown) the five highest-performant models tend to the baseline. This constrasting result indicates that the ensemble mean is less accurate than the hard-to-beat benchmark, though the models have skill at predicting both frequent and infrequent events when all ensembles are considered.

The relative improvement (decrease) of the climatology's Brier Score over the NWPs as the threshold increased is expected,

because the BS is artificially reduced with decreased event frequency as the number of correct rejections grows (Bartels et al., 2017). However, the CRPS, which represents an analysis of all possible thresholds, reveal that all S2S models are skilful against the same baseline. Both the BS and the CRPS evidence that the S2S predictions are generally reliable and discriminatory and have resolution and sharpness at the catchment scale, though skill decreases as the threshold for event





definition increases. Moreover, the FCS at Q70 highlights that the forecasts become more consistent with time, particularly
after two weeks. FCS values at higher thresholds becomes so small that are negligible. This is particularly useful as the interest
in S2S predictions starts after the medium range of predictability.

It is important to note that, apart for ECMWF and ISAC-CNR, most models showed significant bias in predicting both
precipitation and temperature. Indeed, Li et al. (2019) concluded that bias correction significantly improves S2S performance
and should be carried out. In that regard, though the ECMWF and ISAC-CNR models fare particularly well in all evaluated
metrics, literature on the verification of S2S concludes that model quality is very dependent on the studied region (Li et al.,
2019). For example, the RLP and Upper Main sites show no significant difference in skill, however Li et al. (2019) found that
KMA and UKMO performed better than ISAC-CNR in south China.

It can be thus stated that the catchment-level aggregated S2S forecasts have skill at the medium to sub-seasonal range,
meanwhile predictions are then also less likely to fluctuate than at short temporal ranges. This indicates that the S2S can be a
useful product for long-term forecasting high to low frequency events.

## 4.2 Hydrological performance of S2S forecasts at the catchment level

Figure 3 shows the discharge lead-time dependant BS Q90, FCS Q70 and ROCSS Q90 score values for all S2S models and
hydrological benchmarks. The same trend of worsening forecast quality and increasing forecast consistency is observed,
however the yield lead-time is shifted to around 30 to 40 days. Performance at stability is better than or the same as the baseline
for all models for Q90 and for the ECMWF NWP at Q97 (not shown), which also shows notable quality in the meteorological
assessment. The ensemble means of the forecasts also outperform the climatology and the perfect model (the most
discriminating benchmark for hydrology) for RMSE and MAE, thus proving more accurate than the baselines. The FCS is
much larger (higher variability) at short-range for discharge than for precipitation and temperature, however FCS decreases
(consistency increases) significantly up to four weeks, which is within the range of applicability of S2S forecasts. Likewise,
the ROCS highlights that most models have a better hit to false alarm ratio than chance up to 20 days, and thus bring added
value in applications such as flood forecasting and alerting in the medium to long-range scale (e.g. an increase in preparedness).
Therefore, the absolute performance and skill of discharge forecasts forced with S2S predictions is better than those of the raw
forcing due to hydrological memory (or high autocorrelation, see Pappenberger, Thielen and Del Medico, 2011).

The absolute quality of discharge predictions for the Upper Main catchment (averaging over Schwürbitz and Kemmern) is
slightly worse than that of the Schwürbitz sub-catchment, however the Upper Main has more skill in comparison with the
climatology. This is due to area-aggregation effects, where the initial state of a catchment has a bigger influence on the
simulation the larger the area of the catchment is (Pappenberger, Thielen and Del Medico, 2011), and because discharge
timeseries vary more smoothly for more complex river networks (Alfieri et al., 2014).

It is important to note that the data assimilation applied here assumes full knowledge of observations, which artificially
increases the hydrological prediction quality. Regardless, since the rate of decrease in quality is more gradual for discharge
than that for meteorological variables, then the hydrological forecasts remain skilful for longer periods of time.



## 4.3 Model performance predictors

Table 3 shows the feature, score pair and Pearson's R correlation for the significant attributes found, where a negative value means an inverse linear relationship. The correlation analysis evidences that ensemble size is the main predictor of model
performance, appearing as a significant factor for deterministic and probabilistic scores (BS, RMSE, NSE, Value and ROCS), for various threshold levels (Q70 and Q90) and for both the first two weeks (as scores varied with lead-time) and after score stability was reached. Therefore, investments in the explicit definition of uncertainties through ensemble generation pay off, especially for the detection of rarer events (BS at Q90 correlates well with ensemble size). This is in line with recent research on the advantages of ensemble-based over deterministic models (e.g. Bartels et al., 2017; Fan et al., 2015; Richardson, 2000;
Zhu et al., 2002) and with the probabilistic versus deterministic analysis in section 4.1.

Maximum lead-time is overall significant only for lead-times after the yield point. This means that this is likely an artificial consequence of the increase in size of the verification dataset (which is larger for models that have a bigger maximum lead-time). Most importantly, there is no trade-off in performance in the short to medium ranges for models that have bigger maximum lead-times. On the other hand, though original resolution only appears once as significant, it is impossible to discard
it as not important due to model upscaling in the S2S. A verification of the original models must be carried out to better investigate the role of model resolution in the S2S performance. Moreover, an analysis with several more model configurations (not necessarily within the S2S project) would be beneficial for a comprehensive understanding of the other features as well.

## 4.4 Evolution of the lead-time performance of S2S meteorological and hydrological forecasts

Figure 4 shows the composite-model precipitation and discharge lead-time gain from 2015 to 2020 for the Upper Main. The
evaluated score is the Brier Skill Score for Q70, or the Brier Score over the advanced climatology for Q70. On the top the decomposition of gain is plotted: the initial skill is defined as the BSS at the first available timestamp; the gains curve is computed by fitting a regression line through BSS scores at the start and end of the timeseries for each lead-time and then interpolating through a logarithmic function; and the final skill is the sum of the initial plus the gains. The lead-time gain represents the extension of the forecast horizon for a certain skill score: if five days were gained for the 10-day lead-time, it
means that a forecast for 10 days in the future is as skilful in 2020 as a forecast for five days in the beginning of the record.

The evaluation demonstrates that, on average, the S2S forecasts improved over time for both sites. Advancements are larger for smaller threshold levels, while for Q90 and Q97 no betterment is found. It should be noted, however, that there are improvements for the BSS with the persistence benchmark. Still, with the aim of using hard-to-beat benchmarks for a rigorous evaluation of the forecasts, this is not considered here. For precipitation, improvements in the BSS over climatology are smaller
as the lead-time increases but gains in days grows due to the small BSS for longer lead-times. For discharge, gains in the short-range are limited due to the DA methodology used in the initial state definition of hydrological simulations.

Notably, the discharge lead-time improvement in days is dampened in comparison with precipitation, though the discharge forecasts are generally skilful for longer prediction ranges. This is because the skill of the hydrological predictions is generally





higher, thus gains are smaller. Nevertheless, improvements of 5 to 10 days are found and are in line with the results of a large

study of the European Flood Alert System for medium-range forecasts (Pappenberger, Thielen and Del Medico, 2011). By expanding the evaluation range, the current work found that gains in hydrological performance continue for up to at least 30 days.

## 4.5 Correlation between improvements in meteorological and hydrological forecasts

The gains in precipitation lead-time accuracy are plotted against gains in hydrological lead-time from 2015 to 2020 for both

gauges of the Upper Main catchment, as shown in Figure 5. As discussed above, the improvements are much larger for the meteorological forecast than the hydrological ones. Moreover, the advancement in forecast horizon for discharge is unequal due to the complex and non-linear relation between hydrological processes. Here, the hydrological lead-time improvements are twice as big for the smaller sub-catchment than for the whole Upper Main catchment. That is possibly because scores up to Kemmern are better than up to Schwürbitz due to area aggregation effects on hydrological performance, therefore gains

(which are relative to the initial score) are smaller. Indeed, Pappenberger, Thielen and Del Medico (2011) also noted that catchment size has a dampening effect on skill gain over time.

A significant correlation (one that cannot be explained by randomness, with a p-value of $6^{-21}$) was found between advances in the NWPs and their downstream application, as evidenced by the trendlines in Figure 5. The relation is strong up to Schwürbitz ($R^2$ 0.974), though it is also moderate for the whole Upper Main ($R^2$ 0.744). Therefore, the hydrological forecasts performance

does increase proportionally to improvements in the NWPs for the upstream sub-catchment, while area aggregation effects may affect the correlation over whole catchments or bigger sub-catchments.

## 5 Conclusions

We assessed the long-term catchment-averaged performance of nine S2S models regarding nine verification metrics for two highly gauged sites and compared it with the quality of discharge predictions from a coupled hydrological model. Furthermore,

the variation in meteorological and hydrological forecast skill from 2015 to 2020 was verified. The main conclusions from this work are as follows:

- Most S2S models have skill at the catchment-level against the hard-to-beat advanced climatology, particularly for lower threshold levels. Hydrological predictions tend to have more skill at predicting rarer events and to be skillful for longer lead-times (up to 1 month) than meteorological forecasts (around two weeks) due to hydrological inertia.

- Regarding the meteorological quality of the S2S models, the main predictor of forecast skill is ensemble size. Model resolution may also be an important factor, though more work is needed to confirm this hypothesis due to the upscaling of model spatial resolution.





- The NWPs and the resulting hydrological forecasts have, on average, improved from 2015 to 2020 for lower threshold levels. No significant improvement over time has been found for more infrequent events. The gain in lead-time for the same score is higher for meteorological than hydrological predictions, and higher for longer forecast times.

- There is a high correlation between the lead-time gain of meteorological and hydrological performance over time, particularly for smaller catchment areas. This indicates that investments in S2S forecasts increase the forecast horizon of downstream applications.

In view of these findings, this study concludes that forecasts in the sub-seasonal to seasonal range can be valuable to end-users of water resources applications, as it outperforms operationally used base models. Notably, S2S hydro-meteorological systems will become more useful as they improve over time. More research is needed to investigate the role of model resolution in the quality of forecasts, but future development efforts in the S2S should be directed into ensemble generation, as the ensemble size was found to be a significant factor in model performance, particularly for extreme event detection.

**Code and Data Availability**

Code is archived at https://gitlab.com/mbrum/forecast-evaluation; S2S forecasts may be downloaded from the ECMWF portal at https://apps.ecmwf.int/datasets/data/s2s/ (last access: 5 July 2021); discharge observations for the two gauging stations are available from the Gewässerkundlicher Dienst Bayern at https://www.gkd.bayern.de/ (last access: 5 July 2021); meteorological observations and forecasts for Germany are provided by the DWD Climate Data Center at https://cdc.dwd.de/portal/ (last access: 5 July 2021).

**Competing interests**

The authors declare that they have no conflict of interest.

**Author contribution**

MB and DS obtained the data; MB analyzed the data, performed the hydrological modelling, and wrote the manuscript draft; DS reviewed and edited the manuscript.

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




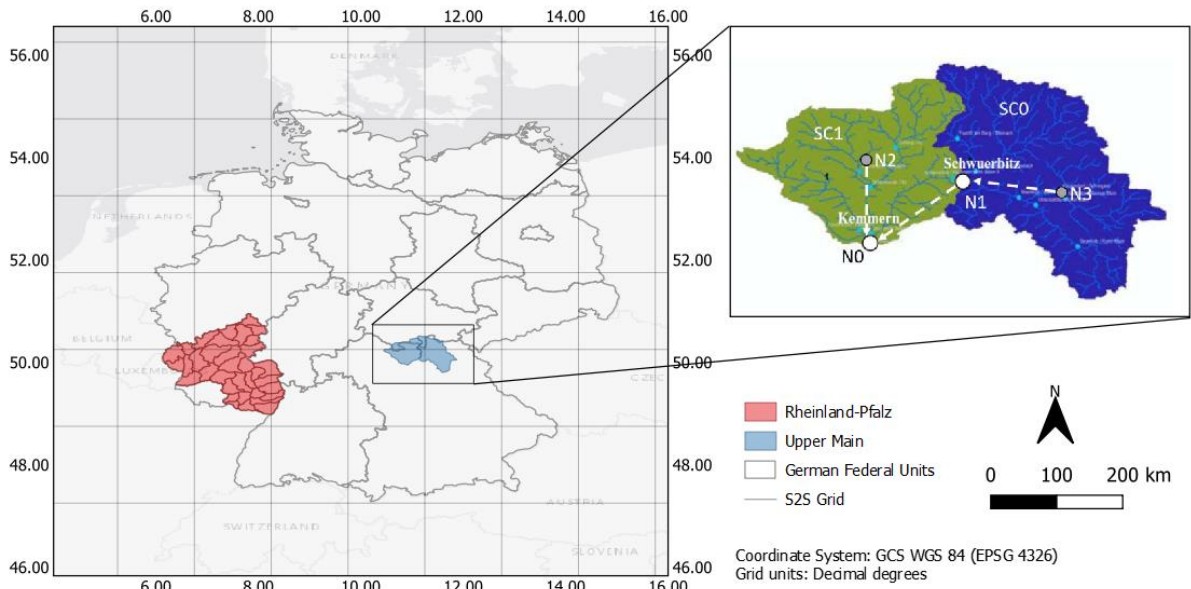

**Figure 1: Location of the two pilot sites and model schematization of the Upper Main catchment. Projection chosen to best represent the S2S grid. Modified from Montero et al. (2016). Background from ESRI.**




| Model | Originating Organization | Code | Maximum Lead-time (days) | Ensemble Size | Original Resolution |
|---|---|---|---|---|---|
| CMA | China Meteorological Administration | babj | 60 | 3 | 0.41°/0.41° |
| ISAC-CNR | National Council of Research of Italy | isac | 32 | 40 | 0.75°/0.56° |
| MF-CNRM | Météo-France | lfpw | 32 | 50 | 0.72°/0.72° |
| ECCC | Environment and Climate Change Canada | cwao | 32 | 21 | 0.45°/0.45° |
| ECMWF | European Centre for Medium-range Weather Forecasts | ecmf | 46 | 50 | 0.14°/0.14° |
| JMA | Japan Meteorological Agency | rjtd | 60 | 49 | 0.36°/0.36° |
| KMA | Korean Meteorological Agency | rksl | 60 | 3 | 0.83°/0.56° |
| UKMO | UK Met Office | egrr | 60 | 3 | 0.83°/0.56° |
| NCEP | USA National Centers for Environmental Prediction | kwbc | 44 | 15 | 0.90°/0.90 |

**Table 1: Characteristics of selected S2S models**



| Score | Equation | Description |
|---|---|---|
| Brier Score (BS) | $$\frac{1}{N}\sum_{i=1}^{N}(y_i - o_i)^2$$ | Probabilistic, threshold-based score. Measures the overall error squared of a probabilistic forecast for a binary event over a verification sample. Assesses reliability, resolution, discrimination, and sharpness of a forecast. |
| Continuous Ranked Probability Score (CRPS) | $$\int_{-\infty}^{+\infty}[P(y) - P_o(y)]^2 dy$$ Where $P(y)$ as the predictive cumulative distribution function and $Po(y)$ the Heaviside step function | Probabilistic, continuous score. Squared difference between the cumulative distribution function of forecasts and observations over the whole spectrum of probabilities. For deterministic forecasts, reduces to the MAE. Assesses reliability, resolution, discrimination, and sharpness of a forecast. |
| Relative Operating Characteristic Score (ROCS) | $$2\sum_{k=1}^{T}\frac{f(F_{k-1}) - f(F_k)}{2}\,\Delta F_k - 1$$ Where $H = f(F)$ is the hit rate, $F$ is the false alarm rate and $T$ is the number of thresholds | Probabilistic, threshold-based score. Measure of potential skill. Compares the difference between the area of a hit rate versus false alarms rate curve (ROC curve) and the area of a no-skill diagonal line. Relates to the capacity of an NWP to issue more correct predictions than false alarms. Assesses the discrimination of a forecast. |
| Relative Economic Value (V) | $$\frac{\min(\alpha,\overline{o}) - F\alpha(1-\overline{o}) + H\overline{o}(1-\alpha) - \overline{o}}{\min(\alpha,\overline{o}) - \overline{o}\alpha}$$ Where $\alpha$ is the cost-loss ratio and $\overline{o}$ the event climatology | Deterministic, threshold-based score. Evaluates the economic value of a forecast depending on the user, model, and the climatology. Compares the expenses averted using a forecast with those from no preventive system (climatology) and benchmarks it against a perfect prediction. |
| Forecast Convergence Score (FCS) | $$\frac{1}{N}\sum_{i=0}^{N}(f_{i,t,l} - f_{i,t-d,l-d})^2$$ Where $f$ is the probability of exceedance for a forecast with lead-time $l$ issued at time $t$, and $d$ is the delay | Probabilistic, threshold-based score. Measures the consistency of a forecast over time. The smaller the FCS, the more consistent a forecast is. |
| Modified Nash-Sutcliffe Efficiency (NSE$_1$) | $$1 - \frac{\sum_{i=1}^{N}|Y_i - O_i|^1}{\sum_{i=1}^{N}|O_i - \overline{O}|^1}$$ | Deterministic, continuous score. First-order measure of overall agreement between predictions and observations. |
| Bias | $$\frac{1}{N}\sum_{i=1}^{N}\overline{Y}_i - \overline{O}_i$$ | Deterministic, continuous score. Measure of systematic difference between forecasts and observations. |
| Root Mean Squared Error (RMSE) | $$\sqrt{\frac{1}{N}\sum_{i=1}^{N}(Y_i - O_i)^2}$$ | Deterministic, continuous score. Measure of average error for a continuous parameter. Purposefully penalizes bigger errors more than smaller deviations. |
| Mean Absolute Error (MAE) | $$\frac{1}{N}\sum_{i=1}^{N}|Y_i - O_i|$$ | Deterministic, continuous score. Measure of average error for a continuous parameter with a focus on the full distribution. More robust than the RMSE. |

$Y$ and $O$ are, respectively, the forecast and observed values; $y$ and $o$ are, respectively, the forecast and observed probabilities; $N$ is the number of samples;

**Table 2: Definition and description of selected evaluation metrics**






**Figure 2: Site-averaged precipitation BS Q70, BS Q90, FCS Q70, CRPS and MAE**




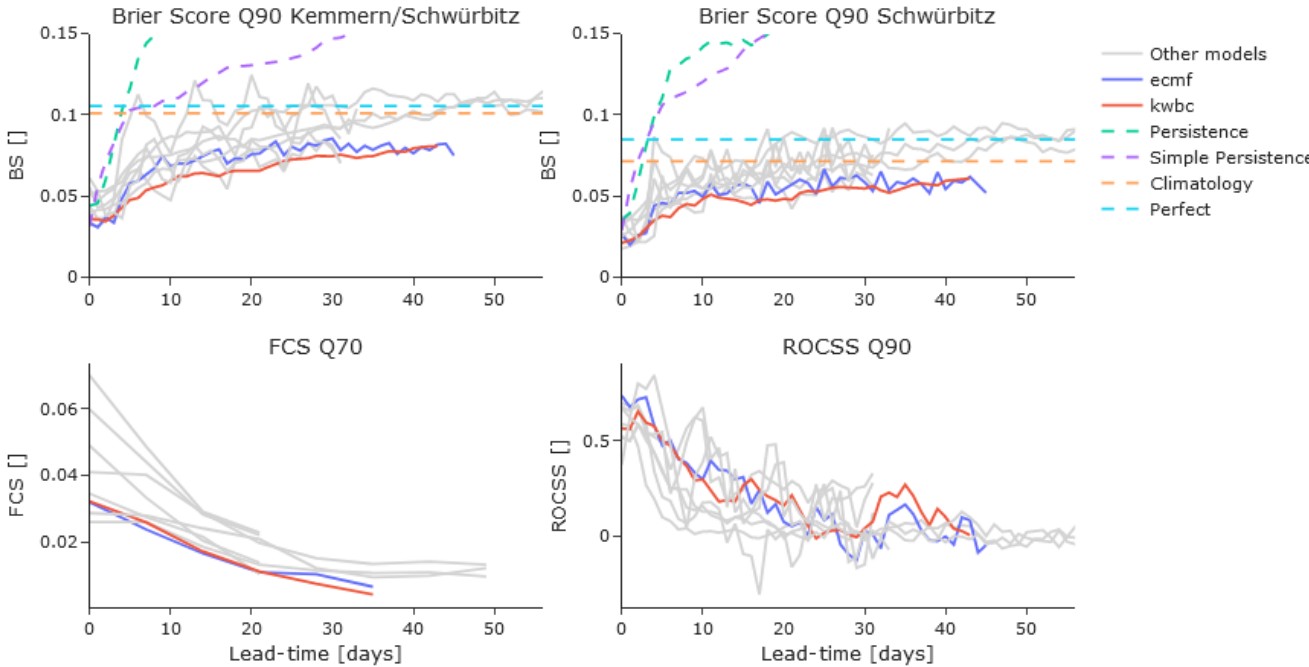

**Figure 3: Discharge BS Q90, FCS Q70 and ROCSS Q90. When not specified, scores are for the whole Upper Main**



| Model Feature | Score | R | Interpretation |
|---|---|---|---|
| Ensemble Size | BS, Q70, First 2 Weeks | -0,714 | Better quality (lower BS) for multiple thresholds with higher number of ensemble members |
| Ensemble Size | BS, Q70, After 2 Weeks | -0,811 | |
| Ensemble Size | BS, Q90, First 2 Weeks | -0,881 | |
| Ensemble Size | BS, Q90, After 2 Weeks | -0,872 | |
| Ensemble Size | RMSE, First 2 Weeks | -0,884 | Better ensemble means accuracy (lower RMSE) with higher number of ensemble members |
| Ensemble Size | RMSE, After 2 Weeks | -0,88 | |
| Ensemble Size | NSE1, First 2 Weeks | 0,854 | Better ensemble means accuracy (higher NSE) with higher number of ensemble members |
| Ensemble Size | NSE1, After 2 Weeks | 0,856 | |
| Ensemble Size | Value, After 2 Weeks | 0,835 | Better economic value (higher Value) at S2S range with higher number of ensemble members |
| Ensemble Size | ROCS Q90, First 2 Weeks | 0,718 | Better forecast discrimination (higher ROCS) with higher number of ensemble members |
| Ensemble Size | ROCS Q90, After 2 Weeks | 0,779 | |
| Maximum Lead-time | BS, Q90; First 2 Weeks | 0,673 | Worse quality (higher BS) for Q90 with bigger maximum lead-time |
| Maximum Lead-time | BS, Q90, After 2 Weeks | 0,68 | |
| Maximum Lead-time | RMSE, After 2 Weeks | 0,723 | Worse ensemble means accuracy (higher RMSE) at S2S range with bigger maximum lead-time |
| Maximum Lead-time | Value Q90, After 2 Weeks | -0,731 | Worse economic value (lower Value) at S2S range with bigger maximum lead-time |
| Maximum Lead-time | ROCS Q90, After 2 Weeks | -0,695 | Worse forecast discrimination (lower ROCS) at S2S range with bigger maximum lead-time |
| Original Resolution | MAE, First 2 Weeks | 0,797 | Worse ensemble means accuracy up to medium range for coarser models |

**Table 3: Significant factor, score combinations for meteorological model performance**




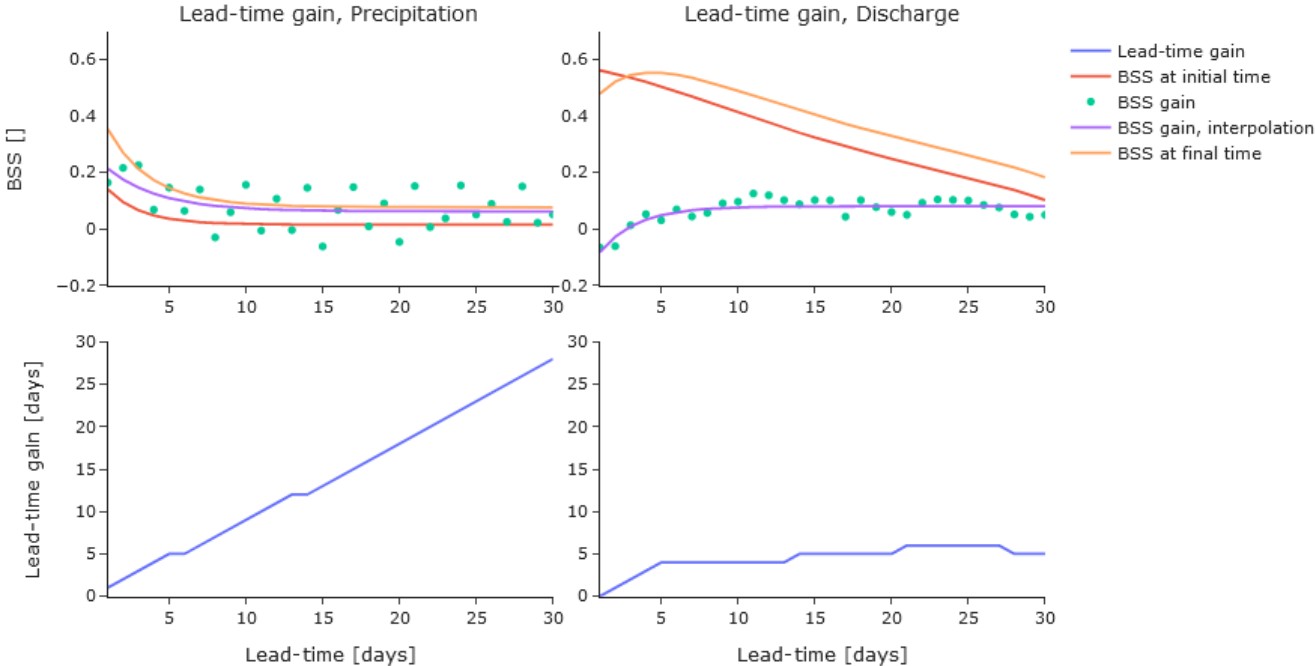

**Figure 4: Precipitation and discharge lead-time gain over time for the whole Up per Main catchment. The lead-time gain is the**
**difference between the lead-time for a score in 2020 (BSS at final time) and the lead-time for the same score at the beginning of the forecast record (BSS at initial time). Graphically, it can be derived by drawing a horizontal line at a given BSS score and taking the difference between the lead-time (x-axis) for the "BSS at final time" and "BSS at initial time" curves.**




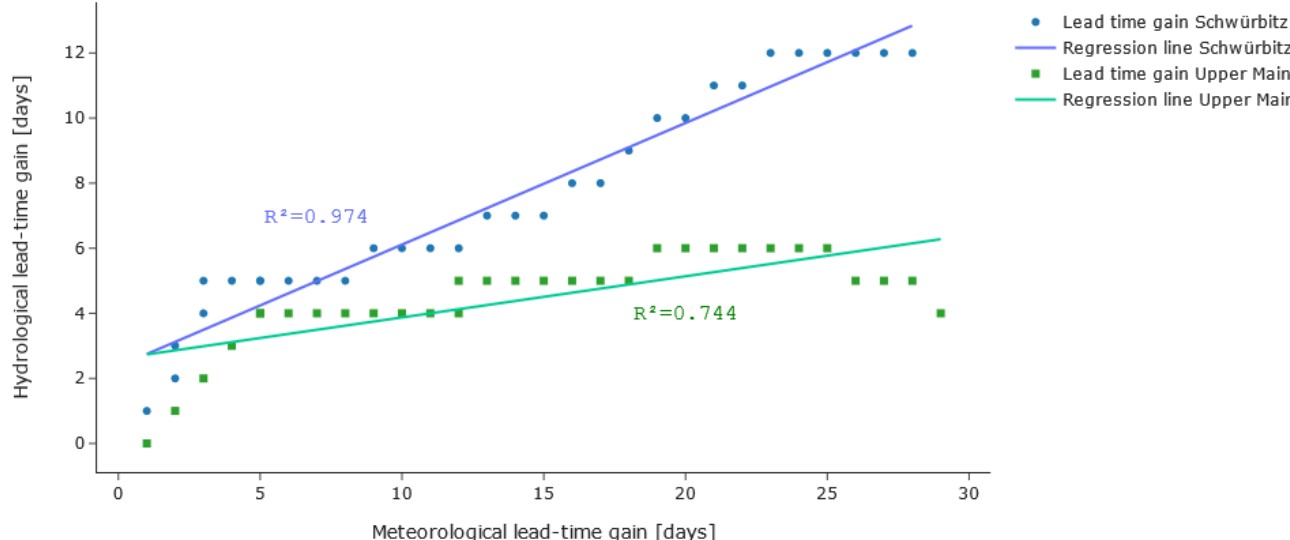


**Figure 5: Correlation between meteorological and hydrological lead-time gain for Schwürbitz and Kemmern (flow at gauge and meteorological results averaged over the whole catchment). The lead-time gain is the difference between the lead-time for a score in 2020 and the lead-time for the same score at the beginning of the forecast record.**