# Peer review of "Long-term evaluation of the Sub-seasonal to Seasonal (S2S) dataset and derived hydrological forecasts at the catchment scale"

_EGUsphere, 2022_

## Author Comment (AC1)

**Summary**

In this study, the authors evaluate the accuracy and hydrologic utility of subseasonal to seasonal (S2S) forecasts generated from the S2S project. The paper is well written and provides a good overview of the performance of the current state-of-the-art S2S forecasts.

However, the paper, as it stands, is lacking in detail, and would especially benefit from better discussion of results.

**Major Comments**

1. **R1: Why were these two study regions selected? The total area is 4100 km2. However, the model grid size is 1.5 x 1.5 deg (~20,000km2) but the authors mention that the study regions encompass 4 grid points. I am not sure how this number was arrived at.**

Authors: The two catchments were selected according to existing models and data, and previous experience of the authors in the areas. Regarding the model grid, the Upper Main catchment spans four grid points, but only partially. The averaging for each parameter takes this into consideration. We have altered the text to clarify this.

Line 72: "… the whole catchment spans four grid points of the S2S raster grid (see the grid delineation in Figure 1)."

2. **R1: I do not fully understand the reason behind not showing the performance of temperature. As NWP modules use different physics for simulating precipitation and temperature, similarity in performance metrics should not preclude the inclusion and discussion of the temperature results as they pertain to different aspects of the NWP model.**

Authors: We have added the temperature results (Figure 3) and discussed them in the text.

[Figure]

Line 195: "Figure 3 presents the same result metrics for temperature. The results for both variables are generally alike, albeit S2S models had constantly better absolute performance for temperature than precipitation, as temperature is generally easier to forecast. Notably, the baseline models are also relatively better at forecasting temperature events, therefore the S2S does not have significantly higher skill for temperature than precipitation. Indeed, skill of temperature predictions in relation to precipitation is highly dependent on the model and inconsistent over the dataset; for each model, this is expected to vary depending on the parameter. Nonetheless, the general yield time of 10 days also holds for temperature."

Line 201: "It is important to note that, apart for ECMWF and ISAC-CNR, most models showed significant bias in predicting both precipitation and temperature, though a positive bias for precipitation was found, while temperatures had to a tendency to be under forecasted."

3. **R1: Why does the performance improve after the yield point? This is very surprising and the authors should provide some explanation as to why performance improves with increase in lead times.**

Authors: Performance after the yield time indeed does not increase, it remains approximately stable, as stated in Chapter 4.1 and 4.2. Noise exists partially because of the different issuance frequency of the S2S models, and the yield point is rather difficult to pinpoint for a few scores (e.g. Bias). Regarding performance improvement over time, it also remains approximately constant after the yield point, as shown in Figure 5.

Line 214: "Interestingly, BS of NCEP for Q70 seems to not stabilize, most likely because it's yield point is above 40 days, which is also seen for two other models with longer maximum lead-times. Persistence and Simple Persistence stabilize at around 50 days and BS scores around 0.2 (not shown in Figure due to scaling)."

4. **R1: The manuscript does not discuss the results comprehensively. What is the impact of ensemble member size on forecast performance? Why do ECMWF models perform better? Have other studies found out the same?**

Authors: We have extended the results chapter thanks to your and Reviewer 2's suggestions. Besides the temperature paragraph (as answered in your second point), we have added comments on the model factor significance and quality evolution over time, along the minor changes requested. I add here those significant improvements:

Line 253: "Though there are other factors to consider, such as the model physics and the region of interest, we can infer that a model with a large ensemble size and high resolution should perform well. Indeed, ECMWF has the biggest number of ensembles and the finest original resolution. Other studies show similar findings: Phakula et al. (2020) found that ECMWF is better at predicting minimum and maximum temperatures than CNRM and UKMO in South Africa; Guimarães et al. (2021) concluded that ECMWF forecasts precipitation anomalies in Brazil better than other S2S models; Deoras et al. (2021) reported that ECMWF has the best ensemble spread-error relationship among all S2S models when predicting Indian monsoon low pressure systems. On the other hand, ISAC-CNR has a higher original resolution and less ensembles (though still more than half of the other models), but its score might be influenced by the small maximum lead-time. As previously discussed, Li et al. (2019) found that KMA and UKMO fared better than ISAC-CNR in south China, which means the good performance of ISAC-CNR might be strongly dependent on the region investigated."

Line 281: "By expanding the evaluation range, the current work found that gains in hydrological performance continue from 14 days to 30 days. It is important to note, however, that the 30 days value is somewhat artificial due to the interpolation methodology and the low averaged BSS scores in 2015. The initial and final BSS curves run parallel from lead-time 14 days on, which means the interpolated final BSS is bigger than any initial BSS, and the computed improvement is consequently constant. Note that computed gain points are very noisy, but do go below the initial BSS. One may thus take conservatively the 14 days as total gain for the system over the 5 years."

Another important point is that due to data availability constraints, the 5 years evaluated are not enough to completely capture the region climatology. This means the improvement seen may be artificially increased by an easy to forecast weather in the more recent years, thus more data would be needed to confirm these findings. Still, the increase in performance over time may be explained by improvements in the model resolution, increases in the number of ensemble members, and changes in parametrization schemes. For example, in March 2017, JMA has increased their model ensemble size from 24 to 49 members; in March 2016 ECMWF doubled their grid resolution; and in July 2019 ECCC upgraded their parameter perturbation methodology. For a complete list of model changes, please see ECMWF (2022). Indeed, we expect that investments in models should result in increased model performance for the meteorological variables produced."

**Minor Comments**

1. **R1: Apart from being a project, S2S is generally used to refer to a specific forecast horizon in the forecasting community. I request the authors to explicitly mention that they are referring to the project.**

Authors: Thank you pointing this out. We have clarified this in the abstract and in the introduction:

Abstract: "Recently, projects such as the Sub-seasonal to Seasonal Prediction Project (S2S)…"

Line 25: "In that regard, recently projects such as the Sub-seasonal to Seasonal Prediction Project (S2S) …"

Moreover, we added an acknowledgement to the project:

L321: "**Acknowledgements**

This work is based on S2S data. S2S is a joint initiative of the World Weather Research Programme (WWRP) and the World Climate Research Programme (WCRP). The original S2S database is hosted at ECMWF as an extension of the TIGGE database."

2. **R1: Abstract: 'Results show that the S2S models have skill at the catchment-scale, particularly for lower threshold levels …'. What does 'lower threshold levels' mean here?**

Authors: We have clarified this point in the abstract:

Line 12: "Results show that the S2S models have skill at the catchment-scale, particularly for less extreme parameter thresholds such as Q70 (30% percentile) …"

3. **R1: Line 30: What is 'hydrological horizon of skillful predictability'?**

Authors: We have changed our wording to improve clarity.

Line 30: "…in the hydrological horizon of skilful predictability (i.e., the maximum lead-time when forecasts are skilful) are…"

4. **R1: ECMWF models are referred to as 'ecmf' and ISAC-CNR as 'isac' in the figures. Please maintain consistency.**

Authors: Those are the model codes, but they are indeed less clear than using the actual model names. We have thus changed the figure legends to use the model names.

---

## Author Comment (AC2)

**Summary:**

The authors present an evaluation of forecast skill of an sub-seasonal 2 seasonal prediction for two regions in Central Germany. The analysis encompasses nine numerical weather prediction (NWPs) models. The authors use a set of well-established forecast skill metrics for this purpose. For both of these regions, the authors analyse meteorological forecast skill for precipitation and temperature. For one of the regions, a hydrologic model (HBV) is applied and forecast skill is evaluated for discharge forecasts. The main findings of the study are that hydrologic forecasts are skillful at longer lead times at which meteorological forecasts are not skillful anymore. The study also shows that forecast have become better over time (i.e., better in 2020 than in 2015) with a larger gain for meteorological variables in comparison to hydrologic variables.

**R2: The nature of the study is more a report than a scientific exploration. For example, the authors provide no explanation why the forecast skill is higher for discharge than for meteorological variables or why the forecast have gained skill over time. The authors could have performed experiments with HBV to show that it is indeed the hydrologic inertia that results in higher skill at longer leads. Similarly, the authors report that forecast gain over time is present, but do not explain why this is happening. Which changes have been applied to the forecast system that could explain the gain. If no change to the forecast system has been applied, this finding might well be an artifact of the experimental design and truly, no gain is present.**

Authors: Regarding possible causes to the increase in forecast skill, we have added a brief discussion on the topic along with some model change examples over time. Within the scope of this paper, we also hope the expansion of the Results and Discussion thanks to your and Reviewer 1's suggestions has improved the quality of the scientific discussion.

Line 253: "Though there are other factors to consider, such as the model physics and the region of interest, we can infer that a model with a large ensemble size and high resolution should perform well. Indeed, ECMWF has the biggest number of ensembles and the finest original resolution. Other studies show similar findings: Phakula et al. (2020) found that ECMWF is better at predicting minimum and maximum temperatures than CNRM and UKMO in South Africa; Guimarães et al. (2021) concluded that ECMWF forecasts precipitation anomalies in Brazil better than other S2S models; Deoras et al. (2021) reported that ECMWF has the best ensemble spread-error relationship among all S2S models when predicting Indian monsoon low pressure systems. On the other hand, ISAC-CNR has a higher original resolution and less ensembles (though still more than half of the other models), but its score might be influenced by the small maximum lead-time. As previously discussed, Li et al. (2019) found that KMA and UKMO fared better than ISAC-CNR in south China, which means the good performance of ISAC-CNR might be strongly dependent on the region investigated."

These substantial points have to be addressed thoroughly before the manuscript can be published. The manuscript is well-structured and overall well written and easy to follow. I only found the figure captions too short and also lacking an introduction of abbreviations used in figures.

**Major comments:**

**R2: The selection of Rhineland-Pfalz (RLP) for evaluating the meteorological performance seems arbitrary. As seasonal forecast are global in nature, this evaluation could be done at that scale. RLP is also very close to the selected catchment (both are located in Germany a few hundred kilometers apart) and share very similar climate. Please consider using another setting for the meteorological evaluation.**

Authors: The two catchments were selected according to existing models and data, and previous experience of the authors in the areas. However, an evaluation of another catchment or at the global level would indeed add value to this study, therefore we have modified the conclusions with an outlook where we address this issue.

Line 334: "Additionally, in future works, we aim to expand this study and investigate the forecast quality of the S2S models applied to catchments with significant different climates and hydrology than the sites evaluated here."

**R2: One of the major points is the gain in forecast performance from the beginning of the forecast period (2015) to the end (2020). 5 years is a very short period that is insufficient to characterize the climatology of the investigated catchment (upper Main in Germany). It is unclear whether the gain in skill is an artefact of catchment observations. In other words, it could well be that the upper Main was easier to forecast in 2020 because of the weather that occured in 2020 than in 2015. The authors need to elaborate on the reasons why they think it is actually an improvement in the system. It is surprising to me that the meteorological lead-time gain can increase by 20 days over such a short period. This means that a forecast at lead time 40 in 2020 is as skillful as a forecast at lead time 20 in 2015. This would be an enormous improvement.**

Authors: The goal here is not to characterize the complete climatology of the region, although we agree that this would be necessary for a complete evaluation. However, we were limited by the 5 years available. Indeed, as you pointed out, the gain in skill over time is both dependent on the actual model improvements and on the ease to forecast the weather. Regarding the large improvement over time, this is partially due to the low score values at the initial reference time, and partially due to the interpolation method used to compute the gains curve. We have expanded the text to better address all these points.

Line 281: "By expanding the evaluation range, the current work found that gains in hydrological performance continue from 14 days to 30 days. It is important to note, however, that the 30 days value is somewhat artificial due to the interpolation methodology and the low averaged BSS scores in 2015. The initial and final BSS curves run parallel from lead-time 14 days on, which means the interpolated final BSS is bigger than any initial BSS, and the computed improvement is consequently constant. Note that computed gain points are very noisy, but do go below the initial BSS. One may thus take conservatively the 14 days as total gain for the system over the 5 years.

Another important point is that due to data availability constraints, the 5 years evaluated are not enough to completely capture the region climatology. This means the improvement seen may be artificially increased by an easy to forecast weather in the more recent years, thus more data would be needed to confirm these findings. Still, the increase in performance over time may be explained by improvements in the model resolution, increases in the number of ensemble members, and changes in parametrization schemes. For example, in March 2017, JMA has increased their model ensemble size from 24 to 49 members; in March 2016 ECMWF doubled

their grid resolution; and in July 2019 ECCC upgraded their parameter perturbation methodology. For a complete list of model changes, please see ECMWF (2022). Indeed, we expect that investments in models should result in increased model performance for the meteorological variables produced."

Line 330: ". To increase confidence in these findings, when data becomes available, a future evaluation over a longer period of time is recommended."

**R2: Table 3: Within this table, only the correlation of forecast feature with the hisghest correlation is shown and others are not reported. It would be necessary to know the correlation of the other model features too to judge how much better the model feature with the highest correlation really is. Additionally, the authors should conduct a statistical significance test to demonstrate that the outperformance by the model features with the highest correlation is significant.**

Authors: Table 3 indeed only shows the significant correlation parameters. As per your suggestion, we carried out a t-test with those features to verify whether they lead to significantly different scores and updated the text and table accordingly. We dropped 5 features which were slightly above to the 0.05 p-value mark, but the overall results remain the same. We have also clarified the table caption. Moreover, for completeness, we have added a table with the complete results (including p-values) to the Supplementary Materials.

Line 125: "We also perform a Student's t-test on the feature-score pairs with significant correlation: we verify whether the models with 50% highest features (e.g., models with more than 21 members) generate significantly different scores than the others."

Line 236: "Table 3 shows the feature, score pair and Pearson's R correlation for the significant (p-value of correlation and p-value of t-test < 0.05) attributes found, where a negative value means an inverse linear relationship. For reference, Table S1 presents the complete significance results including p-values."

**R2: In general, the Figure captions are too short to understand the Figures and need to be expanded. Additionally, abbreviations used in figures are not explained (kwbc in Figure 4) and coloring of lines is also unclear.**

Authors: Thank you for pointing out the lack of clarity in the figure captions. We have changed the captions in the text to improve understanding and altered the figure legends where necessary.

Figure 1: Location of the two pilot sites and model schematization of the Upper Main catchment. S2S raster grid is delineated in light grey over the map. Projection chosen to best represent the S2S grid. Modified from Montero et al. (2016). Background from ESRI.

Figure 2: Site-averaged precipitation BS Q70, BS Q90, FCS Q70, CRPS and MAE per lead-time. The models with best performance overall are highlighted in colour; others are depicted in a grey continuous line for easier visualization. Persistence and Climatology refer to the benchmark models and are shown as dashed lines for comparison.

Figure 3. Site-averaged temperature BS Q70, BS Q90, FCS Q70, CRPS and MAE per lead-time. The models with best performance overall are highlighted in colour; others are depicted in a grey continuous line for easier visualization. Persistence and Climatology refer to the benchmark models and are shown as dashed lines for comparison.

Figure 4: Discharge BS Q90 for Upper Main and Schwürbitz, FCS Q70 and ROCSS Q90 for Upper Main. The models with best performance overall are highlighted in colour; others are depicted in a grey continuous line for easier visualization. Persistence, Simple Persistence, Climatology and Perfect refer to the benchmark models and are shown as dashed lines for comparison.

Table 1: Significant (p-value of correlation and p-value of t-test < 0.05) model feature and average score per lead-time pairs for meteorological model performance. The correlation between feature and score is also added, as well as an interpretation of the combination of model features and metrics.

Figure 5: Precipitation and discharge lead-time gain over time for the whole Upper Main catchment. Above, the decomposition of the average score gain into "BSS at initial time" and "BSS at final time". The "BSS gain" curve is an interpolation of the computed values for each lead-time. Below, the gain in days per lead-time for each parameter. The lead-time gain is the difference between the lead-time for a score value in 2020 and the lead-time for the same score value at the beginning of the forecast record.

Figure 6: Correlation between meteorological and hydrological lead-time gain for Schwürbitz (in blue) and Upper Main (flow at gauge and meteorological results averaged over the whole catchment, in green). Regression lines and their $R^2$ values are also plotted. The lead-time gain is the difference between the lead-time for a score in 2020 and the lead-time for the same score at the beginning of the forecast record.

**Minor edits:**

*Abstract*

**R2: - should mention Germany as location of catchments**

Authors: Mentioned, thank you for pointing this out.

Line 10: "…from a coupled hydrological model for two catchments in Germany."

**R2: - should provide reference to S2S project**

Authors: Referenced, thank you for pointing this out.

Moreover, we added an acknowledgement to the project:

L321: "**Acknowledgements**

This work is based on S2S data. S2S is a joint initiative of the World Weather Research Programme (WWRP) and the World Climate Research Programme (WCRP). The original S2S database is hosted at ECMWF as an extension of the TIGGE database."

*Intro:*

**R2: - paragraph starting at line 48 can be merged with paragraph ending at line 33. They cover the same points.**

Authors: Merged, thank you for pointing this out.

**R2: - line 54: which time? the past decades or lead time.**

Authors: This was indeed the study time, or 5 days. We have changed the text to reflect this.

Line 53: "…over the S2S project duration…"

**R2: - line 56: what do the authors mean by "definite model predictors"?**

Authors: We mean the predictors of model quality. Changed, thank you for pointing this out.

Line 55: "… over the S2S project duration…"

**R2: - line 58: What do the authors verfiy here? The sentence seems to be incorrect. Verify is also not an appropriate word in the context of modelling. Maybe better choose validate.**

Authors: Changed to validate, thank you for pointing this out.

*Materials and Methods*

**R2: - line 74: please modify sentence to make clear which statistic corresponds to which gauge**

Authors: Clarified, thank you for pointing this out.

Line 74: "Catchment-averaged precipitation ranges from 550 to 1,000 mm/year in Schwürbitz and from 650 to 1150 mm/year in Kemmern. Yearly temperatures vary from 6°C to 9°C in both sub-catchments (derived from Montero et al., 2016)."

**R2: - line 98: it is not clear how temperature dataset has been interpolated from the stations to the grid scale.**

Authors: The temperature observations are not interpolated, rather averaged over the whole sub-catchment. We changed the text to better express this approach:

Line 108: Station-based temperature observations are averaged over the stations within the evaluated regions.

**R2: - line 125ff. Some metrics that are introduced here are not used in the results section, for example 'value'. These should be removed.**

Authors: Value is indeed not extensively discussed, but it is addressed in the model features section, where we find that Maximum Lead-time is an explanatory factor for Value. Therefore, we maintain in the table.

**R2: - line 155: HBV uses a triangular weighting function for the river routing. It is not clear to me how this is used to connect the nodes N0, N1, N2, and N3. It needs to be**

**clarified          what          these          nodes          represent.**

Authors: Thank you for your comment. We have added a sentence to clarify.

Line 158: "We apply the rainfall-runoff routine to each of the 30 sub-basins and generate daily values, which are added at nodes N2 and N3 and then routed to the outflow stations N0 and N1 through a simple, non-linear reservoir routing scheme."

**R2: - Line 157: It is not clear to me how a data assimilation (DA) is run for HBV. HBV is a conceptual hydrologic model where model state variables cannot be mapped to observation-based variables. Please expand. It seems like DA improved the performance (see line 166ff), but the model performance even without DA is very high and certainly within the range deemed as useful for end users. Please clarify why DA is necessary at all for this application.**

Authors: The DA here was carried out to minimize model uncertainty during the performance evaluation of the NWPs. The original DA implementation was carried out by Schwanenberg and Montero (2016). We added these clarifications to the text.

Line 161: "In addition, we perform a variational data assimilation (DA) procedure following Schwanenberg and Montero (2016) to receive a best guess of the system state at forecast time in each sub-catchment and minimize model uncertainty during the performance evaluation of the NWPs."

*Results:*

**R2: Line 178: it should be stated that Q97 is not shown**

Authors: Stated, thank you for pointing this out.

Line 217: "… at Q97 (not shown) …"

**R2: Line 180: The sentence starting at 'Persistence, on the...' is not clear to me. Could you please                                                                                                   rephrase.**

Authors: Thank you for your comment. We rephrased it.

Line 182: "The persistence benchmark, on the other hand, is an unreliable baseline for skill discrimation, because it is outperformed by all the evaluated models."

**R2: Line 198: please state that this statement is only valid for the meteorological forecasts.**

Authors: Stated, thank you for pointing this out. We modified the text as follows:

Line 208: "It can be thus stated that the catchment-level aggregated S2S meteorological forecasts have skill at the medium to sub-seasonal range…"

**R2: Figure 3: it is confusing that the name for both gauges contain Schwuerbitz in the name.**

Authors: We have changed the name of the Kemmern/Schwürbitz gauge to Upper Main.

**R2: Figure 3: kwbc is not explained**

Authors: We changed the legend from the model codes to the model names for clarity. "kwbc" means "NCEP".

**R2: Line 239ff: It is unclear to me how the composite-model lead-time gain is calculated.**

Authors: We have re-worded the calculation sentence for clarity.

Line 265: "Given a lead-time value and its model-averaged BSS score at the final evaluation time, the gains equal the difference between the lead-time in days minus the lead-time of the BSS same score at the start of the timeseries. On the top the decomposition of gain is plotted: the initial skill is defined as the BSS at the first available timestamp; the gains curve is computed by fitting a logarithmic function to the gains; and the final skill is the sum of the initial BSS plus the gains"